# Gut-Faecal Microbial and Health-Marker Response to Dietary Fumonisins in Weaned Pigs

**DOI:** 10.3390/toxins15050328

**Published:** 2023-05-11

**Authors:** Yarsmin Yunus Zeebone, Brigitta Bóta, Veronika Halas, Balázs Libisch, Ferenc Olasz, Péter Papp, Tibor Keresztény, Annamária Gerőcs, Omeralfaroug Ali, Melinda Kovács, András Szabó

**Affiliations:** 1Agribiotechnology and Precision Breeding for Food Security National Laboratory, Institute of Physiology and Nutrition, Hungarian University of Agriculture and Life Sciences, Guba S. Str., H-7400 Kaposvár, Hungary; 2ELKH-MATE Mycotoxins in the Food Chain Research Group, Hungarian University of Agriculture and Life Sciences, Guba S. Str., H-7400 Kaposvár, Hungary; 3Department of Farm Animal Nutrition, Institute of Animal Physiology and Nutrition, Kaposvár Campus, Hungarian University of Agriculture and Life Sciences, Guba S. Str., H-7400 Kaposvár, Hungary; 4Agribiotechnology and Precision Breeding for Food Security National Laboratory, Institute of Genetics and Biotechnology, Hungarian University of Agriculture and Life Sciences, Szent-Györgyi. Str., H-2100 Gödöllő, Hungary; 5Doctoral School of Biology, Hungarian University of Agriculture and Life Sciences, H-2100 Gödöllő, Hungary; 6Doctoral School of Biology, ELTE Eötvös Loránd University, H-1117 Budapest, Hungary

**Keywords:** mycotoxin, *Fusarium*, fumonisin B, weaned pigs, gut microbiota, faecal microbiota, health-markers

## Abstract

This study investigated effects of dietary fumonisins (FBs) on gut and faecal microbiota of weaned pigs. In total, 18 7-week-old male pigs were fed either 0, 15 or 30 mg FBs (FB_1_ + FB_2_ + FB_3_)/kg diet for 21 days. The microbiota was analysed with amplicon sequencing of the 16S rRNA gene V3-V4 regions (Illumina MiSeq). Results showed no treatment effect (*p* > 0.05) on growth performance, serum reduced glutathione, glutathione peroxidase and malondialdehyde. FBs increased serum aspartate transaminase, gamma glutamyl-transferase and alkaline phosphatase activities. A 30 mg/kg FBs treatment shifted microbial population in the duodenum and ileum to lower levels (compared to control (*p* < 0.05)) of the families *Campylobacteraceae* and *Clostridiaceae*, respectively, as well as the genera *Alloprevotella*, *Campylobacter* and *Lachnospiraceae Incertae Sedis* (duodenum), *Turicibacter* (jejunum), and *Clostridium sensu stricto 1* (ileum). Faecal microbiota had higher levels of the *Erysipelotrichaceae* and *Ruminococcaceae* families and *Solobacterium*, *Faecalibacterium*, *Anaerofilum*, *Ruminococcus*, *Subdoligranulum*, *Pseudobutyrivibrio*, *Coprococcus* and *Roseburia* genera in the 30 mg/kg FBs compared to control and/or to the 15 mg/kg FBs diets. *Lactobacillus* was more abundant in the duodenum compared to faeces in all treatment groups (*p* < 0.01). Overall, the 30 mg/kg FBs diet altered the pig gut microbiota without suppressing animal growth performance.

## 1. Introduction

The *Fusarium verticillioides* and *F. proliferatum* fungi are primarily responsible for producing the fumonisin B series (FBs), a major mycotoxin group of toxicological importance. The most common food/feed commodities that are contaminated with FBs are maize and products made of maize [1,2]. Fumonisins- B_1_ (FB_1_), B_2_ (FB_2_), B_3_ (FB_3_) and B_4_ (FB_4_) are the most common types of FBs in feedstuffs, with proportions of approximately 70–80%, 15–25%, 3–8% and 1–2%, respectively, in most field samples [3]. The most toxic of these subspecies is FB_1_ and has been the subject of extensive research [4]. The structural analogy of FB_1_ to sphingosine (So, i.e., the major long-chain base backbone of cellular sphingolipids) has been established as a competitive inhibitor of sphinganine (Sa) and the enzyme sphingosine N-acyltransferase—also known as ceramide synthase (CerS)—has been established as the principal pathway of FB_1_-induced toxicity in most examined species [5]. This enzyme suppression disrupts sphingolipid metabolism, resulting in increased Sa and So in the serum and tissues of animals and a reduction in complex sphingolipids. Thus, the Sa/So ratio has been used across several animal species as an early biomarker of FB_1_-induced toxicity [5]. Additionally, clinical signs induced by FBs are species-specific and vary depending on the primary target organ [5].

FB_1_ has been shown to have adverse effects including equine leukoencephalomalacia (ELEM) in horses, porcine pulmonary edema (PPE) and hydrothorax in pigs [6], as well as hepatotoxic effects/cancer [7,8] and nephrotoxicity in rodents [9]. Oesophageal cancer and neural tube anomalies in humans have been connected to consuming foods contaminated with FB_1_ in several parts of the world [10,11]. According to the International Agency for Research on Cancer (IARC), FBs (FB_1_ + FB_2_) are classified as a category 2B carcinogen [12]. Frequently, FBs toxicity studies focus on the post-absorptive effect, which tends to be a lapse in their impact on the gastrointestinal tract (GIT) environment. In monogastric animal species like the pig, FBs absorption rate is only 1 to 6 per cent [13]. Together with the enterohepatic recycling role of the gut, the GIT is recurrently exposed to much higher levels of FBs relative to other tissues [14]. Some studies have evaluated the potentially deleterious effects of FBs on GIT functionality and have reported immunomodulation [15], intestinal barrier dysfunctions [13] and a reduction in nutritional digestion and absorption efficiency in the gut [16,17,18]. Unfortunately, the potential effects of FBs toxicity on the other major coordinator of GIT functioning—the microbiota—has received little attention.

According to Conway [19], pig gut health is primarily determined by how well the diet, mucosa and commensal flora are all working. The strong relationship between mucus and microflora implies that the detrimental effects caused by FBs on immune functions ultimately cause changes in the composition and structure of the gut microflora [20]. Disruption of this triple balance can lead to dysbiosis and disease in animals (for review, see: [21,22]). Thus, fundamental importance of the gut microbiota is demonstrated by the numerous health benefits this complex and dynamic ecosystem provides to its hosts to improve the host’s health. Studies on mycotoxins, including those on aflatoxin B_1_ (AFB_1_) [23], zearalenone (ZEN) [24] and trichothecenes (TCT) [25] have revealed negative effects on the composition of the microbiota. In a few studies, the presence of FB_1_ (purified form; 0.5 mg/kg body weight, BW) in the diets acted as a predisposing factor for the increased invasion of *Escherichia coli* colonization in the intestines of piglets [26]. Burel and others [27] found that continuous exposure to 11.8 ppm FBs (FB_1_ + FB_2_) reduced the faecal microbiota profiles, which resulted in an imbalance of the microbiota of pigs. Zhang et al. [28] highlighted a significant shift in the microbial flora of BALB/c mice exposed to increasing levels of dietary FBs, while Yu et al. [29] found that a significant shift in the microbiota of broilers exposed to either 10 or 20 mg/kg FBs (FB_1_ + FB_2_ + FB_3_) or their hydrolysed from for eight weeks had an imbalance in their intestinal microbiota, as well as growth retardation and tissue damage. 

Furthermore, research on mycotoxins and the microbiome frequently focuses on the faecal microbiota with only occasional consideration of the small intestinal microbiota. It should be noted, though, that many dietary components are broken down and absorbed in the small intestine. As a result, the GI epithelium is exposed to a variety of foreign substances, including mycotoxins and microbial components from the diet. Therefore, it is essential to investigate the negative effects of mycotoxins in the small intestine.

In the current study, we examined how dietary FBs (FB_1_ + FB_2_ + FB_3_) at two dietary levels of 15 or 30 mg/kg FBs fed to weaned pigs for 21 days affected several health markers and the microbiota composition of the small intestine and faeces of weaned pigs. 

## 2. Results

### 2.1. Growth Rate and Clinical Signs

The current experiment was designed to study other GIT processes such as nutrient digestibility and mineral retention and as such, the growth rate and feed intake parameters have been reported already in a companion paper [30]. One important factor is to note that the dosages of dietary FBs (15 and 30 mg/kg FBs diet) used in the experiment were designed to exceed the EU guidance level of FBs in complete feed for pigs (5 mg/kg FBs diet) to trigger the toxic effects for the 21-day duration of the study. 

No growth impairment, morbidity or mortality was found in all groups during the 21-day trial. Even though brief episodes of diarrhoea were seen during the two-week acclimatization period, the pigs were fully recovered before the trial started [30]. 

### 2.2. Serum Biochemical Endpoints

Results on serum biochemical parameters as a response to different levels of dietary FBs are summarized in Table 1. The blood total protein concentration was significantly increased (*p* = 0.012) in G3 as compared to G1 and G2. Aspartate transaminase (AST), gamma glutamyl-transferase (GGT) and alkaline phosphatase (ALKP) activities were all significantly elevated (*p* < 0.05) in G3 compared to G1; G2 was indifferent regarding GGT and ALKP levels. Creatinine kinase (CK) was significantly elevated (*p* = 0.019): about 3-fold in G3 compared to G1 or G2. There was also a notable rise (*p* = 0.003) in the lactate dehydrogenase (LDH) activity in G3 compared to G1 or G2. Feeding pigs with a 30 mg/kg FBs diet significantly elevated (*p* < 0.05) creatinine and cholesterol concentrations, as well as calcium (Ca) and magnesium (Mg) levels compared to G1 or G2. When compared to the other two groups, G3 had a significantly declined glomerular filtration rate (GFR; *p* = 0.029). However, there was no dietary FBs influence (*p* > 0.05) on the serum levels of albumin, alanine transaminase (ALT), urea, iron (Fe), sodium (Na) and chloride (Cl) in all groups.

### 2.3. Antioxidant Parameters 

None of the antioxidant parameters—glutathione (GSH) or glutathione peroxidase (GSHPx)—nor the end-product of lipid peroxidation—malondialdehyde (MDA)—determined in the lung, liver, kidney, or plasma exhibited any significant alterations in their levels (*p* > 0.05) of pigs receiving either a control, 15 or 30 mg FBs/kg diets for 21 days (Appendix A).

### 2.4. Histopathological Findings 

Histopathological alterations in the liver, kidney, lung, and small intestine of the pigs in all groups (G1, G2 and G3) are detailed in Appendix A through to Appendix A. There were no recognizable histopathological alterations examined in the liver of control animals (G1). Most developed histopathological alteration was in the liver of G3 where multifocal lysis, necrosis of hepatocytes and multifocal swelling and proliferation of mononuclear phagocyte system (MPS)-cells of the liver are visible (Appendix A). 

In Table 2, histopathological alteration scoring results of the kidney, lung and ileum are presented. Following a thorough examination of each animal in each group, the major pathological alterations were described and assessed for severity and extent as follows:
0 = no alteration, healthy condition;1 = slight/small scale/few extent of pathological alteration;2 = medium degree/medium scale/medium number of pathological modification;3 = pronounced/extensive/numerous extent or occurrence of the alterations.

The total number of animals exhibiting pathological symptoms according to the scoring described above are summarized and presented in Table 2. In the three examined segments of the small intestine (duodenum, jejunum, and ileum), only ileum showed alteration, such as lymphocyte depletion in the gut-associated lymphoid tissue. All six animals fed either 15 or 30 mg/kg FBs diet exhibited all the symptoms associated with liver damage compared to the control animals.

### 2.5. Microbiota Analyses 

Due to the highly complex nature of the gut microbiota, our analysis focused on identifying bacterial taxa (families or genera) with an altered relative abundance following FBs treatment. Figure 1 shows the phylum-level mean bacterial composition in the examined pig intestinal sections of the three treatment groups (G1, G2 and G3). The Firmicutes was the most abundant phylum in all intestinal sections and in faeces. In the faeces of day 0, Firmicutes and Bacteroidetes dominated and were followed by the Proteobacteria and Spirochaetae. In the faeces of day 21, Firmicutes increased (*p* = 0.03) while Bacteroidetes declined (*p* = 0.03) compared to day 0. Overall, in all examined intestinal sections, Firmicutes dominated, where in the duodenum (G1d), Firmicutes had 84.9% mean relative abundance while Actinobacteria and Proteobacteria followed in close succession with 5.8% and 4.8%, respectively. In the jejunum (G1j), Tenericutes was the second most abundant phylum (12.46%), followed by Actinobacteria and Proteobacteria (2.87% and 1.24%, respectively). In the ileum (G1i) however, Proteobacteria (3.1%) was the second most abundant, followed by Tenericutes (2.2%) (Figure 1).

Phylum-level mean relative abundances within the same intestinal sections and between treatment groups were not significantly different (*p* > 0.05). However, as mentioned earlier, our analysis focused on identifying bacterial taxa that were differentially abundant after FBs treatment. Overall, between the 30 mg FBs/kg diet (G3 group) and between the control (G1) and/or the 15 mg FBs/kg diet (G2 group), a total of 13 significant changes were found at the genus level (Table 3). The faecal samples showed the highest number of significant differences (*p* < 0.05) in the relative abundances of eight genera. In the duodenum, jejunum, and ileum; three, one and one genera/genus showed a statistically altered (increased or decreased) abundance in response to FBs treatment, respectively (Table 3). In the duodenum, *Campylobacter* (G1 vs. G3, G2 vs. G3 comparison; *p* = 0.016, 0.012) (Figure 2), *Alloprevotella* (G1 vs. G3 comparison; *p* = 0.013) (Figure 3) and *Lachnospiraceae: Incertae Sedis* (G1 vs. G3; G2 vs. G3 comparison *p* = 0.009, 0.035, respectively) showed a significant reduction. Likewise, *Turicibacter* in the jejunum and *Clostridium sensu stricto 1*. in the ileum were both decreased in a G1 vs. G3 comparison with *p*-values of 0.001 and 0.009, respectively. In the faecal samples, there was a significant increase in the relative abundances of the genera *Solobacterium*, *Faecalibacterium*, *Anaerofilum*, *Ruminococcus*, *Subdoligranulum*, *Pseudobutyrivibrio*, *Coprococcus* and *Roseburia* in a G1 vs. G3 and/or in a G2 vs. G3 comparison (*p* < 0.05) (Table 3).

Significant changes in the mean relative abundances of genera listed in Table 3 between the three treatments are provided in Appendix A, together with comparisons of the observed number of species (Appendix A) and the inverse Simpson index values (Appendix A) between the three treatments, as calculated by the QIIME pipeline. The observed number of species and inverse Simpson index values did not display a normal distribution by the Shapiro-Wilk test and were therefore further analysed by the non-parametric Kruskal–Wallis test. Analyses performed by Kruskal–Wallis tests did not demonstrate significant differences between treatments within the same intestinal sections (*p* > 0.05) for the observed number of species and inverse Simpson index values.

Although *Lactobacillus* relative abundances were not significantly different (*p* > 0.05) between the three treatment groups within a particular intestinal segment (Figure 4), Kruskal–Wallis tests demonstrated that *Lactobacillus* abundances were markedly higher (*p* < 0.01) in the proximal part of the intestinal tract (that is, in the duodenum) compared to the faeces in all treatment groups. Moreover, Pearson correlation analyses showed that *Lactobacillus* relative abundances correlated between the duodenum and ileum (r = 0.86, *p* < 0.01), and between the ileum and faeces (r = 0.67, *p* < 0.05) among the three examined groups (Figure 4).

## 3. Discussion

In this study, male weaned pigs were exposed to either 15 or 30 mg/kg dietary FBs for 21 days to examine some health indicators and the modifying effect on the gut and faecal microbiota distribution.

Following 21 days of dietary exposure of FBs, the weaned pigs exhibited no growth retardation or morbidity. Notes garnered in the literature on the effects of FBs on the growth performance of various animal species are described as inconsistent and loss of body weight (BW) is not always the case. Oftentimes, FBs toxicity in pigs only interferes with pigs’ growth performance at doses above 100 mg FB_1_/kg feed and effects range from four to eight weeks whereas lower doses only slightly cause devastation or none [31] which is consistent with our findings. Furthermore, in a recent study where piglets were fed concentrations ranging from low to high i.e., 3.7, 8.1 or 12.2 mg FB_1_/kg diet for 28 days, the authors reported no performance alteration of piglets [32]. 

In pigs, the primary target organs of FB_1_ are the liver, kidney and lungs, and organ-specific serum biochemical endpoints provide sensitive response [33], and histopathological findings are typically concomitant. The effects of FBs on modifying blood biochemical parameters and suggesting internal organ damage are frequently dose-dependent [34,35]. In the present results, 30 mg FBs/kg feed resulted in a significant elevation of the liver enzyme aspartate transaminase (AST) (about 5 folds increase as compared to the control group)—an increment that exceeds the normal physiological range of 31 to 58 U/L [36]. The histopathology revealed notable damage in the liver indicative of hepatocyte death. In addition, the significant increase in serum cholesterol levels corroborates the revelation of liver damage reported previously [37,38]. After 8 days of feeding 5 mg FBs/kg BW, Dilkin et al. [37] reported a notable rise in blood cholesterol. In addition to a significant elevation in blood cholesterol in the work of Schertz et al. [35], the authors reported no FBs related impact on serum alkaline phosphatase (ALKP) and AST levels after acute oral exposure of barrows to 3425 nmol FB_1_/kg BW, thus partially agreeing with our findings. 

In the kidney, alterations were pronounced and frequent in animals receiving the 30 mg/kg FBs diet. These alterations were indicative of progression of renal toxicity and confirmed by the significant decline in serum GFR, elevation in CK, and creatinine levels in the 30 mg/kg FBs fed pigs. These alterations in the kidney may have shifted the levels of the electrolytes (Mg and Ca) as well. Moreover, despite the supposition that FBs may induce oxidative damaging effects on the liver, kidneys, and biological fluids, none of the antioxidant parameters (GSH, GSHPx) nor the lipid peroxidation end-product (MDA) examined appeared to be affected by dietary FBs. However, the Hsp70 expression of the liver of pigs fed the 30 mg/kg FBs diet was remarkably elevated as already reported in a companion paper [30]. In slight agreement with the present outcome, Kócsó et al. [39] reported an elevated expression of Hsp70 activity in the kidney and lung, but not the antioxidant parameters GSH, GSHPx or MDA of rats fed with a 50 mg/kg FB_1_ in the diet for 5 days. Given these observations, it is possible that a factor unrelated to the mitochondrial respiratory chain triggered the rise in Hsp70 production. 

The efficient functioning of the gut microbiome is governed by the interplay of intrinsic and extrinsic factors (physiological state of the animal, feed and nutrients availability and their endogenous secretion into the gut lumen, immune status, housing, environmental conditions, etc.) [23,40,41,42]. Consistent with the literature, the present study found that the *Firmicutes* and *Bacteroidetes* were the most common phyla found in all the pigs’ intestinal tracts (Figure 1). Earlier works with pigs reported similar findings [43,44,45]; a similar trend in humans [46] and in broilers [29]. The reduction in the relative abundance of *Alloprevotella* (Figure 3) belonging to the *Prevotellaceae* family and order *Bacteroidales* is in line with the conclusions reached by Zhang et al. [28]. Additionally, in an 8- week- long study involving BALB/c mice intragastrically exposed to much lower doses of 0.162, 0.486, 1.458, and 4.374 mg/kg FB_1_, the authors reported a marked reduction in *Alloprevotella* [28]. Due to its capacity to create succinate and acetate i.e., two compounds with anti-inflammatory characteristics and roles in strengthening the intestinal barrier [47], this Gram-negative bacterium is considered as beneficial [48]. In addition, in a G1 vs. G3 comparison, there was a significant decrease for the *Lachnospiraceae*-*Incertae sedis* group (Table 3) and for the order *Clostridiales* in the duodenum and in the ileum, respectively. Butyric acid is produced by these bacteria according to reports [49,50], and its profound benefits include reducing intestinal inflammation and fortifying the intestine’s response to dietary changes [51]. Given the outcome of the present study, it is conceivable that the presence of FBs in pigs’ diet has a detrimental effect on the proliferation of some essential intestinal bacteria involved in gut barrier fortifying responses. Additionally, these observed alterations in the relative abundances of several crucial bacteria found in the intestines reinforce the notion that the gut is highly susceptible to damage from dietary FBs. 

The faecal bacterial flora after 21 days showed a dominance of the families *Ruminococcaceae* (14.1%) and *Lachnospiraceae* (17.8%). Further, the differentially abundant genera related to these two prominent families were *Solobacterium*, *Faecalibacterium*, *Anaerofilum*, *Ruminococcus*, *Subdoligranulum*, *Pseudobutyrivibrio*, *Coprococcus* and *Roseburia* (Table 3). According to research, healthy piglets’ guts were comprised more of *Prevotellaceae*, *Lachnospiraceae*, *Ruminococcaceae* and *Lactobacillaceae* when compared to diarrheic piglets [52]. A similar pattern was observed when pigs were fed a diet containing 12 mg/kg FB_1_ for 0, 8, 15, 22 and 29 days. The abundance of the families *Lactobacillaceae*, *Lachnospiraceae*, *Ruminococcaceae* and *Prevotellaceae* were all substantial, but *Lachnospiraceae* experienced a considerable reduction through time [53]. In response to exposure by 12 mg/kg FBs diet, Mateos et al. [53] reported a significant increase of OTUs assigned to *Prevotella*, *Treponema* and *Lactobacillus*, and a significant decrease in the relative abundances of OTUs belonging to *Faecalibacterium*, *Prevotella*, *Mitsuokella*, *Roseburia*, *Ruminococcus* and *Succinivibrio*. Intriguingly, several of these latter genera also had a tendency for a decreased relative abundance (*p* > 0.05) in a G1 vs. G2 comparison in our work, where using a similar FBs dose in G2 (15 mg FBs/kg feed) to that of Mateos et al. (12 mg FB_1_/kg diet). For *Prevotella*, this decrease was also significant in the current G1 vs. G2 comparison (*p* = 0.037), like Mateos et al. [53]. 

In another study, Burel et al. [27] used the single-strand conformation polymorphism (SSCP) faecal microbiota profiles and reported a temporary imbalance of the microbiota of pigs chronically fed with an 11.8 ppm FB_1_ diet during the first four weeks of exposure and then fluctuating until finally becoming similar again at the end of the trial. In addition, when the GIT was colonized with *Salmonella*, the result was more dire [27]. Thus, an indication that a slight or minor change in the microbiota paves way for the possible invasion of opportunistic pathogenic bacteria. 

The genus *Lactobacillus* is the most prevalent genus in lactic acid bacteria, and the most predominant genus in the small intestine [54]. The increasing trend in *Lactobacillus* sp. that was seen in the present study in a dose-response-like pattern (Figure 4), though statistically not significant (*p* > 0.05), may have been brought on by a gut microbial response to counteract any potential negative effects of the presence of FBs in the small intestine. The high relative abundance of *Lactobacillaceae* in the various sections of the small intestine—duodenum, jejunum and ileum—has also been reported elsewhere [55]. Similar to the current investigation, Moon et al. [56] subjected pigs to the *Fusarium* mycotoxin deoxynivalenol (DON; 0.8 mg/kg) for 30 days and found a considerably higher relative abundance of *Lactobacillaceae* in the small intestine as opposed to the faeces. In fact, it seems that *Lactobacillus* thrives most abundantly in the small intestine in response to FBs or DON exposure. Additionally, we discovered a striking decrease in the *Campylobacter* genus (Figure 2), a member of the family *Campylobacteraceae*, which has been linked to post-weaning diarrhoea in piglets [21]. Possibly, a covert factor in suppressing the growth of *Campylobacter* notably in the duodenum of the treated pigs was the action of *Lactobacillaceae*. Even though this is just a hypothesis, it all serves to demonstrate how the microbiota composition can be altered while attempting to balance any possible adverse effects following exposure to FBs, and thereby indicating an immense significance for their hosts. Likewise, similar to the current study, Moon et al. [56] reported that the administration of DON contaminated feed to pigs resulted in an altered composition of the gut microbiota (where *Lactobacillaceae* also dominated the small intestine) even though the animals did not develop any typical clinical signs. 

Increased shifts in *Lactobacillus* sp. following mycotoxins including FBs exposure have been emphasized in other investigations as well. In their study, Mateos et al. [53] reported a remarkable increase of *Lactobacillus* which was also noted in an in vitro investigation to determine how FB_1_ interacts with pig caecal microorganisms [57]. In the work of Dang et al. [57], *Lactobacillus* and total bacteria increased, while anaerobic bacteria showed a considerable reduction. Elsewhere, after exposing both nursery and growing pigs to multi-toxins including FBs, there was an increase in the *Lactobacillacae* family in the intestinal microbiome [58]. These outcomes confirm the establishment that *Lactobacillus* sp. can reduce mycotoxin toxic activities by the extra-cellular binding of mycotoxins [59,60]. The fundamental mechanism was explained as a process of physical adsorption involving distinctive constituents of the cell wall [61]. Our current observations thus further support the need for future investigations using various *Lactobacillus* strains that may be isolated and possibly employed as probiotics to enhance gut health in the advent of fumonisin toxicosis. 

Because of their capacity to complete tasks that individual strains or species cannot, microbial consortia have attracted increasing attention for their use in toxic substances biodegradation [62]. The current investigation identified three bacterial genera in the pigs’ gut—*Pseudomonas* (of the family *Pseudomonadaceae* and order Pseudomonadales), *Sphingomonas* (of the family *Sphingomonadaceae* and order Sphingomonadales), and *Achromobacter*—that may be able to break down FBs. *Pseudomonas* showed an abundance trend of G1 < G3 in all examined intestinal sections with a tendency for higher levels in the proximal part of the intestine; *Sphingomonas* showed a notable abundance in the duodenum and jejunum of the G3 group only, and lastly, *Achromobacter* was most abundant also in the duodenum and jejunum of the G3 group (Appendix A), respectively, are shown in Appendix A. Even while these differences were not statistically significant (*p* > 0.05), it is plausible that an intestinal consortium of microorganisms may act together to break down FBs also in the intestine. In a partial agreement, from used mushroom compost, Zhao et al. [63] identified the SAAS79 FB_1_ degrading bacterial consortium, which primarily included members of the *Pseudomonas*, *Comamonas*, *Delftia*, *Sphingobacterium* and *Achromobacter* genera. The authors found that with a 90% degradation rate, SAAS79 could degrade FB_1_ in 3 h into less harmful products and attributed this to a possible synergistic interaction between the bacterial consortia’s species in the degradation process owing to the consortium’s lack of any active single degraders [63].

## 4. Conclusions

The current study examined the compositional effects of dietary FBs on the gut and faecal microbiota of weaned pigs, as well as some health indicators. Dietary treatments resulted in a significant decrease of some beneficial bacteria such as *Alloprevetolla* and *Lachnospiraceae*: *Incertae Sedis* (in the duodenum), *Turicibacter* (in the jejunum) and *Clostridium sensu stricto 1* (in the ileum). Using 30 mg FBs/kg feed the faecal microbiota shifted to a higher relative abundance for some gut-health-promoting families such as the *Ruminococcaceae* and at the genus level for *Solobacterium*, *Faecalibacterium*, *Anaerofilum*, *Ruminococcus*, *Subdoligranulum*, *Pseudobutyrivibrio*, *Coprococcus* and *Roseburia*. Although statistically not significant, a trend for increasing mean relative abundances of *Lactobacillus* was observed between the control (G1), G2 and G3 treatment groups, similar to the study of Moon et al. [56], where in response to DON contaminated feed, an altered gut microbiota (dominated by the *Lactobacillaceae* in the small intestine) was reported, without any typical clinical signs. Overall, the findings of the current study indicate that fumonisin B series mycotoxins can interfere with and modify the composition of intestinal and faecal microbiota in young pigs without necessarily negatively impacting animal performance at an already severe hepatotoxic and nephrotoxic state (AST, LDH, CK, total cholesterol, liver histopathology revealing cellular damage in a dose dependent occurrence). Furthermore, the *Lactobacillus* relative abundance patterns observed in the gut of FBs or DON exposed pigs provide further support for future investigations on the detoxification of fumonisin B by certain *Lactobacillus* strains of probiotic potential.

## 5. Materials and Methods

The experiment was carried out according to the regulations of the Hungarian Animal Protection Act. The allowance number for the studies was SOI/31/00308-10/2017 (date of approval: 28 February 2017, by the Hungarian National Scientific Ethical Committee on Animal Experimentation and issued on 27 March 2017 by the Somogy County Government Office, Department of Food Chain Safety and Animal Health).

### 5.1. Animals, Housing and Experimental Diets 

The current work is a part of a larger investigation into how the functioning of the GIT and certain tissue phospholipids’ fatty acid profiles responded to dietary FBs intoxication in weaned pigs. As such, animal care and preparation of diets, animal euthanasia and sample collection herein described have been previously reported in three separate companion papers [30,64,65].

After a two-week physiological acclimation phase to establish the gut microbiota, 18 male Danbred weaned pigs, averaging 13.5 ± 1.3 kg, were enrolled in the study, and assigned to one of three diets at seven weeks of age. The control group was fed a diet that contained no fungal culture of FBs, while the treatment groups were fed with 15 mg/kg, or 30 mg/kg FBs containing diet for 21 days. Feed was offered as an amount that covers 2.5 times the maintenance energy requirement and was provided twice a day, in two equal portions. A corn-soybean-based diet of commercial origin was used as the basal diet. Unconsumed feed was measured back every day. Table 4 shows the proximate nutrient content of the feed given. Drinking water was made available *ad libitum* [30,64,65].

Mycotoxin contamination and experimental diets preparation have already been reported [64]. In brief, the fungal strain *Fusarium verticillioides* (MRC 826) was inoculated on pre-soaked, sterile maize kernels, in a form of spore suspension. Fungal culture was produced according to [66]. The final FBs (FB_1_ + FB_2_ + FB_3_) concentrations were 2000–4000 mg/kg in the air-dried culture material harvested in different batches. The fungal culture was mixed into the ration of the experimental animals to provide feed concentration of 15 and 30 mg/kg daily FBs. 

From the feed lots (100 kg/lot), 12 samples totalling about 80 g were randomly selected and blended into bigger lot samples. The lot samples were finely ground, and the mycotoxin concentration of representative analytical subsamples was determined. The diet fed to the control group did not contain detectable amounts of FBs. In the diets, the absence of FBs co-occurrence with DON, ZEN, and T-2 toxin was also tested and excluded, in which the analysed diets did not contain detectable concentrations (limits of detection, LODs were 0.053, 0.005, and 0.011 mg/kg for DON, ZEN and T-2 toxin, respectively) [64].

Concentrations of mycotoxins in prepared samples were determined with a Shimadzu 2020 liquid chromatography-mass spectrometry (LCMS) system (Shimadzu, Kyoto, Japan). To obtain high-resolution chromatographic separation, a XB-C18 Kinetex analytical column (100 × 2.1 mm, 2.6 µm; Phenomenex, Torrance, CA, USA) was used with a 0.25 mL/min flow rate (injected sample volume: 10 µL). The gradient elution was performed with eluents A (0.2% formic acid + 0.005 M ammonium formate) and B (methanol + 0.005 M ammonium formate), using the following gradient programme: 0.0–1.0 min 10 % eluent B, 1.0–13.0 min linear increase of eluent B to 100 %, 13.0–16.0 min 100 % eluent B, 16.0–17.0 linear decrease of eluent B to 10 %, and 17.0–20.0 min 10 % eluent B. Three different mass per charge ratio (*m*/*z*) values were used for each mycotoxin (i.e., 1 for quantification and 2 for confirmation of the detected mycotoxin) as shown in Table 5 (ELKH-MATE Mycotoxins in the Food Chain Research Lab’s own method, unpublished), and the mycotoxins concentration used for the experimental diets follows in Table 6. 

### 5.2. Sampling and Laboratory Analysis

#### 5.2.1. Blood Serum Biochemistry

The concentration of plasma total protein (TP), albumin, creatinine concentration, the activities of alanine aminotransferase (ALT), aspartate aminotransferase (AST), gamma-glutamyl transferase (GGT), lactate dehydrogenase (LDH) and alkaline phosphatase (ALKP) were determined using Roche Hitachi 912 Chemistry Analyzer (Hitachi, Tokyo, Japan) and commercial diagnostic reagent kits (Diagnosticum Ltd., Budapest, Hungary). Glomerular filtration ratio was calculated by the analyser software using creatinine as basic data.

#### 5.2.2. Determination of Antioxidant Parameters and Lipid Peroxidation

For the determination of the antioxidant parameters and lipid peroxidation, kidney, liver, and lung samples were stored at −80 °C until analysis. Lipid peroxidation was determined by the quantification of malondialdehyde (MDA) levels with a 2-thiobarbituric acid method in cell hemolysate [67]. The concentration of reduced glutathione (GSH) was measured by the method of Sedlak and Lindsay [68], and the activity of glutathione peroxidase (GSPHx) according to Lawrence and Burk [69]. 

#### 5.2.3. Histology and Histopathology

Following macroscopic external and internal examination of the organs, the liver, kidney, lung, and small intestine were fixed in 10% buffered formaldehyde and embedded in paraffin. Sections of 5 µm thickness were stained with haematoxylin-eosin (H.E.) and examined by light microscopy. Individual animals from each group were examined and the main pathological alterations were described and scored according to the extent and severity as follows:
0 = no alteration, healthy condition;1 = slight/small scale/few extent of pathological alteration;2 = medium degree/medium scale/medium number of pathological modification;3 = pronounced/extensive/numerous extent or occurrence of the alterations.

These examinations were performed by Autopsy Public Claims Company Limited, Budapest, Hungary.

#### 5.2.4. Statistical Analysis 

Statistical analyses of the antioxidant parameters and serum biochemical measurements data were performed using the SPSS version 20.0 software (IBM Corp., Armonk, NY, USA). Results were subjected to a one-way analysis of variance (ANOVA) and in a case of significant treatment effect, the Tukey post-hoc test was used to check the intergroup differences. A *p*-value of < 0.05 was considered significant. 

#### 5.2.5. Microbiota Analysis and Statistical Analysis Procedure 

For the intestinal microbiota analysis, the pig intestine was placed on a sterile autoclave bag immediately after dissection of the animals, on which each intestinal section was excised. Prior to excision, 10 cm intestinal sections were sealed at both ends of the region with sterile bundles and the sealed intestinal section was placed in a sterile Petri dish. At the next stage of sampling, one end of the intestinal section was cut with a sterile scalpel and the contents were transferred into a sterile urine collection vessel and evenly homogenized with a sterile spatula. From each of the homogenized samples, approximately 0.2 g aliquots were measured with a sterile spatula into four (4) sterile 1.5 mL Eppendorf tubes. Four (4) samples from each intestinal section were stored as duplicates for metagenomic use. The samples in the Eppendorf tubes and the samples remaining in the urine storage vessel were immediately placed in a freezer at −20 °C and stored at −70 °C for long periods after sampling was completed. The faecal samples were collected into 500 mL volume sterile polyethylene bags which were fixed on peri-anal silicone plates attached to the piglets. Samples for the metagenomic analysis were collected on the morning of the day of slaughter and were handled and sub-sampled the same way as the samples from the other intestinal sections.

DNA purification from 100-200 mg of each intestinal content sample was performed by LGC Genomics GmbH (Berlin, Germany) according to their DNA extraction service from stool samples to yield 20–200 ng of genomic DNA, dissolved in Tris/TE (5 mM, pH: 8.5) with the concentration of ≥1–10 ng/µL. Amplicon sequencing of the V3–V4 region of the 16S rRNA gene was performed by LGC Genomics GmbH (Berlin, Germany). Library preparation and sequencing were performed using an Illumina MiSeq platform with v3 chemistry. DNA fragments were amplified using amplification primers 341F (5′-CCTACGGGNGGCWGCAG-3′) and 785R (5′-GACTACHVGGGTATCTAATCC-3′) [70]. Primers also contained the Illumina sequencing adapter sequence and a unique barcode index. Resulting amplicons were sequenced using the Illumina MiSeq v3 600-cycle kit to provide paired-end read lengths of 2 × 300 bp. Demultiplexing of all libraries for each sequencing lane were attained using the Illumina bcl2fastq 2.17.1.14 software (Illumina Inc., San Diego, CA, USA). Combination of forward and reverse reads were carried out using the BBMerge 34.48 tool [71]. The mothur software package (v1.35.1, [72]) was used for pre-processing and operational taxonomic units (OTUs) picking from Illumina amplicon sequencing data by clustering at the 97% identity level.

Creation of relative abundance taxonomical tables in an Excel format was performed with QIIME 1.9.0. (LGC Genomics GmbH, Berlin, Germany) [73]. Relative abundance data generally did not follow a normal distribution, as assessed by the Shapiro–Wilk test [74]. Inverse Simpson index analyses and observed species number calculations were performed on rarefied data by QIIME at the sampling depth of 21727 reads per sample. Differential abundance testing in the duodenum, jejunum and ileum content, and faeces of the three treatment groups (Control (G1), 15 mg/kg FBs (G2) and 30 mg/kg FBs (G3)) was performed by the non-parametric Kruskal–Wallis test using IBM SPSS Statistics 27.0 software (SPSS Inc., Chicago, IL, USA) [75,76,77]. A difference was considered significant at *p* < 0.05. The bacterial composition of day 0 and day 21 faecal samples in the three treatment groups was compared using the Wilcoxon signed-rank test [74,75,78].

## Figures and Tables

**Figure 1 toxins-15-00328-f001:**
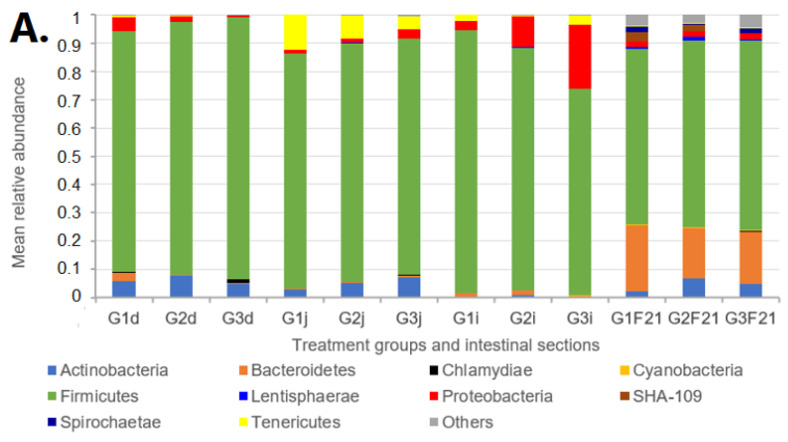
(**A**) Phylum-level bacterial composition of the examined pig intestinal sections: duodenum (d), jejunum (j), ileum (i) and day 21 faeces (F21) in the three treatment groups (G1, G2 and G3), where data show mean relative abundances of the six animals analysed in a group. (**B**) Phylum-level bacterial composition of day 0 faeces (F0) and day 21 faeces (F21) in the three treatment groups (G1, G2 and G3), where data shown are mean relative abundances of the six animals analysed in a group.

**Figure 2 toxins-15-00328-f002:**
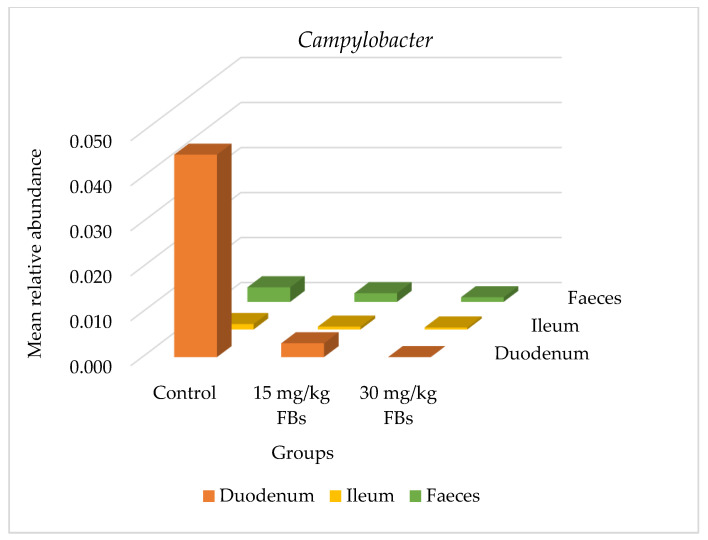
Bar graph showing the mean relative abundance of *Campylobacter* in the duodenum, ileum, and faecal microbiota of weaned pigs (data are mean of 6 pigs/group).

**Figure 3 toxins-15-00328-f003:**
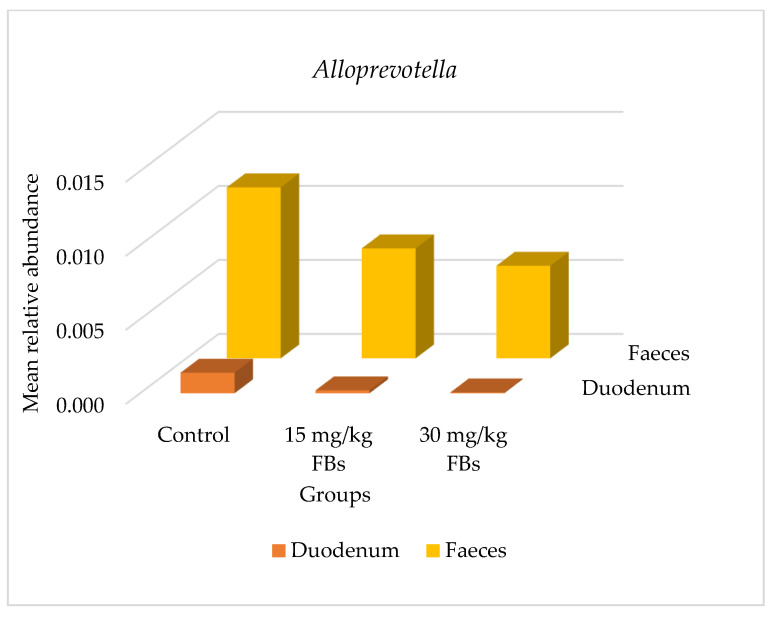
Bar graph showing the mean relative abundance of *Alloprevotella* in the duodenum and faecal microbiota of weaned pigs (data are mean of 6 pigs/group).

**Figure 4 toxins-15-00328-f004:**
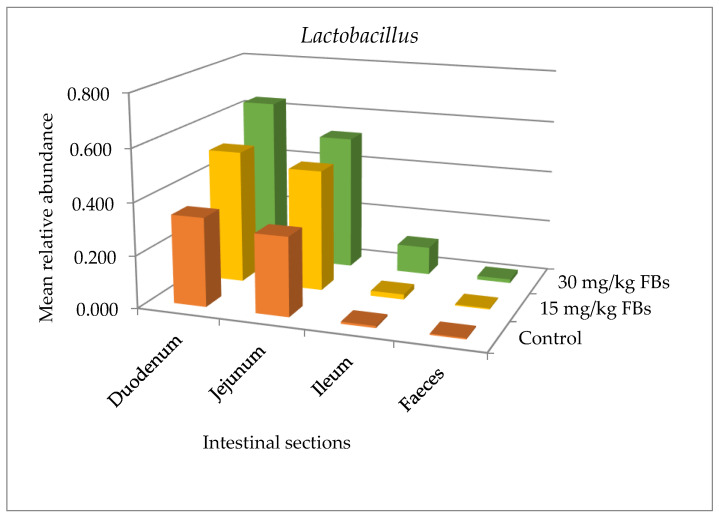
Mean relative abundance of *Lactobacillus* in the gut and faecal microbiota of weaned pigs (data are mean of 6 pigs/group).

**Table 1 toxins-15-00328-t001:** Effects of dietary FBs on serum biochemical parameters of weaned pigs (data are means ± standard deviation (SD) of 6 individuals/group).

Blood Serum Parameters	Control (G1)	15 mg/kg FBs (G2)	30 mg/kg FBs (G3)	*p*-Value
Total protein, g/L	56.2 ± 2.7 ^a^	56.8 ± 2.2 ^a^	60.9 ± 2.8 ^b^	0.012
Albumin, g/L	35.6 ± 3.6	34.4 ± 1.8	35.7 ± 3.8	0.703
AST, U/L	57.5 ± 18.2 ^a^	87.8 ± 22.9 ^a^	336.5 ± 269.6 ^b^	0.001
ALT, U/L	65.7 ± 16.7	86.3 ± 22.89	100.8 ± 55.3	0.117
GGT, U/L	37 ± 7.0 ^a^	53.2 ± 18.5 ^ab^	122.8 ± 102.2 ^b^	0.011
ALKP, U/L	270.7 ± 40.6 ^a^	406 ± 212.3 ^ab^	1356.5 ± 1384.6 ^b^	0.008
CK, U/L	1382.5 ± 686.5 ^a^	1351.5 ± 852.2 ^a^	4449.8 ± 3315.5 ^b^	0.019
LDH, U/L	1192 ± 165.7 ^a^	1323.8 ± 212.7 ^a^	2272.3 ± 693.7 ^b^	0.003
GFR, ml/min	90 ± 0.0 ^a^	88.7 ± 3.3 ^a^	76.4 ± 12.6 ^b^	0.029
Urea, mmol/L	3.3 ± 0.6	4.3 ± 0.9	3.9 ± 0.1	0.212
Cholesterol, mmol/L	2.2 ± 0.1 ^a^	3 ± 0.5 ^a^	4.5 ± 0.9 ^b^	0.001
Creatinine, µmol/L	88 ± 8.17 ^a^	88 ± 8.17 ^a^	102.3 ± 4.5 ^b^	0.019
Ca, mmol/L	2.7 ± 0.13 ^a^	2.7 ± 0.18 ^ab^	2.9 ± 0.11 ^b^	0.029
Mg, mmol/L	1.0 ± 0.12 ^a^	1.0 ± 0.10 ^a^	1.2 ± 0.12 ^b^	0.018
Fe, mmol/L	26.1 ± 4.8	21.7 ± 3.4	24.1 ± 6.7	0.353
Na, mmol/L	147.3 ± 4.1	147 ± 2.6	143.5 ± 3.0	0.116
Cl, mmol/L	101.5 ± 5.7	116.8 ± 41.3	97.3 ± 2.9	0.183

ALKP = alkaline phosphatase, ALT = alanine transaminase, AST = aspartate transaminase, CK = creatine kinase, GFR = glomerular filtration rate, GGT = gamma-glutamyl transferase, LDH = lactate dehydrogenase. Different superscripts within a row indicate a significant difference at *p* < 0.05.

**Table 2 toxins-15-00328-t002:** Histopathological alterations (occurrence frequency) observed in animals (6 pigs/group) fed either a control diet (G1), and 15 or 30 mg/kg FBs diets (G2 and G3, respectively) for 21 days.

Organ Pathological Observation	Control (G1)	15 mg/kg FBs (G2)	30 mg/kg FBs (G3)	
	n			∑
Liver				
Decrease of staining intensity of liver cells	0	6	6	12
Single liver cell death	0	6	6	12
Swelling of MPS cells	0	6	6	12
Proliferation of MPS cells	0	6	6	12
Kidney				
Tubular epithelium detachment	0	0	6	6
Lymphocytic infiltration	1	1	6	8
Lung				
Interstitial lymphocytic infiltration	2	3	4	9
Pleural fibrosis	0	2	3	5
Ileum				
GALT lymphocyte depletion	0	0	3	3

GALT = gut-associated lymphoid tissue, MPS = mononuclear phagocyte system, n = number of animals in each group exhibiting an examination symptom, ∑ = the total number of animals exhibiting the given alteration irrespective of dietary FBs treatment.

**Table 3 toxins-15-00328-t003:** Colonization pattern for differentially abundant genera observed in animals (6 pigs/group) fed either a control diet (G1), and 15 or 30 mg FBs/kg diets (G2 and G3, respectively) for 21 days.

Sample Origin	Genus-Level Colonization Pattern	Group Effects	*p*-Value
Duodenum	*Bacteroidales: Prevotellaceae: Alloprevotella*	G1G3 ↓	0.013
Duodenum	*Campylobacteraceae: Campylobacter*	G1G3, G2G3 ↓	0.016, 0.012
Duodenum	*Firmicutes*: *Clostridiales: Lachnospiraceae: Incertae Sedis*	G1G3, G2G3 ↓	0.009, 0.035
Jejunum	*Firmicutes*: *Erysipelotrichaceae*: *Turicibacter*	G1G3 ↓	0.001
Ileum	*Clostridiales: Clostridiaceae: Clostridium sensu stricto 1.*	G1G3 ↓	0.009
Faeces	*Firmicutes: Erysipelotrichaceae: Solobacterium*	G1G3, G2G3 ↑	0.040, 0.003
Faeces	*Firmicutes: Clostridiales: Ruminococcaceae: Faecalibacterium*	G2G3 ↑	0.020
Faeces	*Firmicutes: Clostridiales: Ruminococcaceae: Anaerofilum*	G1G3, G2G3 ↑	0.027, 0.008
Faeces	*Firmicutes: Clostridiales: Ruminococcaceae: Ruminococcus*	G2G3 ↑	0.011
Faeces	*Firmicutes: Clostridiales: Ruminococcaceae: Subdoligranulum*	G2G3 ↑	0.015
Faeces	*Firmicutes: Clostridiales: Lachnospiraceae: Pseudobutyrivibrio*	G1G3, G2G3 ↑	0.031, 0.027
Faeces	*Firmicutes: Clostridiales: Lachnospiraceae: Coprococcus*	G2G3 ↑	0.04
Faeces	*Firmicutes: Clostridiales: Lachnospiraceae: Roseburia*	G2G3 ↑	0.011

Symbols are the following: ↑ indicates a significant increase, ↓ indicates a significant decrease between the groups, G1 = control group, G2 = 15 mg/kg FBs dose group, G3 = 30 mg/kg FBs dose group. A difference in relative abundance values was considered significant at *p* < 0.05. See further details in Appendix A.

**Table 4 toxins-15-00328-t004:** Analysed nutrient content of the experimental feed.

*Item*, g/kg	
Crude protein	175
Crude fat	33
Crude fibre	37
Crude ash	50
Starch	418
Lysine	11.1
Methionine	3.7
Ca	6.5
P	5.0
*Item*, mg/kg	
Manganese	40
Zinc	110
Iron	87
Copper	9.4
Selenium	0.30
Iodine	0.6
DE, MJ/kg	14.70
ME, MJ/kg	14.10

**Table 5 toxins-15-00328-t005:** Mass per charge ratio (*m*/*z*) values used for five different mycotoxins in LCMS system.

Mycotoxins	Quantification (*m*/*z*)	Confirmation (*m*/*z*)
FB_1_	722.4 (+)	760.3 (+), 720.3 (−)
FB_2_	706.4 (+)	744.3 (+), 704.4 (−)
DON	335.0 (+)	297.0 (+), 340.9 (−)
ZEN	317.0 (−)	357.0 (+), 319.0 (+)
T-2 toxin	505.1 (+)	484.2 (+), 589.0 (+)

(+) or (−) indicates polarity of the ionic charge, DON = deoxynivalenol, ZEN = zearalenone.

**Table 6 toxins-15-00328-t006:** Level of total fumonisins used for the formulation of contaminated diet, and limit of detection of fumonisins and other mycotoxins in experimental feed.

Mycotoxins	LOD, mg/kg	Control (G1)	15 mg/kg FBs Diet (G2)	30 mg/kg FBs Diet (G3)
FB_1_	0.031	nd	15.4 *	29.75 *
FB_2_	0.051	nd		
FB_3_	-	nd		
DON	0.053	nd	nd	nd
ZEN	0.005	nd	nd	nd
T-2 toxin	0.011	nd	nd	nd

* = FB_1_ + FB_2_ + FB3, FB_1_ = fumonisin B_1_, FB_2_ = fumonisin B_2_, FB_3_ = fumonisin B_3_, DON = deoxynivalenol, LOD = limit of detection, nd = not detected, ZEN = zearalenone.

## Data Availability

Not applicable.

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
