# Peer review of "Gut-Faecal Microbial and Health-Marker Response to Dietary Fumonisins in Weaned Pigs"

_toxins, 2023, doi:10.3390/toxins15050328_

Round 1
Reviewer 1 Report
I have reviewed the review entitled “Gut-faecal microbial and health-marker response to dietary fumonisins in weaned pigs” (toxins-2254954).
After my review of the manuscript, it is my judgment that it is a well-designed work. I only have several minor suggestions.
1. References should be added in Lines 44-48.
2. Line 77, as we know that DON is one kind of trichothecenes. So, the statement is illogical, and I suggest the authors to re-write the sentence.
3. For the Figure 1, I suggest the authors to re-prepare a new one to replace the current Fig 1-b.
4. The use of ‘sp.’ such as in Lines 355 and 363 are wrong, it should not be italic.
5. Line 431, ‘Fusarium verticillioides’ should be italic.
6. There should be a space between the number and the unit. For example, in Line 447, etc.
7. For the units of volumes, different forms were used in the main text, such as ‘ml, mL, μl, and μL’. I suggest the authors to use the unified format, especially for the same unit.
8. The use of ‘x’ in several places are wrong (Lines 502, 536, etc.).
9. I suggest the authors to double check the sentences between Lines 521-525.
10. Line 549, ‘P’ should be italic.
11. I suggest the authors to re-write the ‘Conclusions’, some speculative analysis should be put into the discussion section, such as the sentences between Lines 396-400. What’s the highlights or significance of the research? It’s better to be mentioned in the ‘Conclusions’.
Reviewer 2 Report
Review on toxins-2254954: Gut-faecal microbial and health-marker response to dietary fumonisins in weaned pigs
The authors intended to investigate the impact of dietary fumonisin on the gastrointestinal tract with special emphasis on the luminal microbial community in pigs. In view of the climate changes and the consequences for mycotoxin occurrence this is surely commendable.
However, the manuscript shows serious flaws that require attention before it is fit for publication. I’d recommend to address all the issues and then re-submit this manuscript.
Material & Methods:
As part of a larger project the authors cited previous publications for a detailed description of Material & Methods (reference 38: Zeebone et al., J Anim Physiol Anim Nutr. 2022;1–14, DOI: 10.1111/jpn.13724). Unfortunately, the prerequisite of the study, i.e. the production of the fumonisin-contaminated diets, does not match between this manuscript and the previous publication: analysis showed 15.40 and 29.75 mg FUM/kg diet in the already published article, i.e. FUM comprises FB1+FB2+FB3, whereas in this manuscript Table 6 details the same concentrations for FB2 alone. Which figure is correct?
Furthermore, no information is provided as to the mode of dietary sampling in order to measure mycotoxins. As mycotoxins are usually present rather as “traces” in the diet, the mode of sampling directly impacts the quality of analysis.
Histology & Histopathology: there appears to be an unnecessary section concerning the assessment of small intestinal architecture = villi & crypts (lines 475-484). This section is the exact same text as in the above-mentioned previous article and the results were presented there. Please remove this section!
Lines 485-495 pertains to the histological assessment of liver, kidney, lung and ileum and does not provide an actual scoring protocol as to what exactly was indeed viewed per organ and coherently determined. There are a number of published histopathological scores and just stating various degrees of alterations from none to severe is not enough. Please provide a proper table on the actual tissue and cell-based analysis done microscopically. In the results section one might gather some assessments but this should be properly provided in this section.
SCFA: Please provide a rationale as to why you only sampled cecal content for analysis of SCFA, whereas for your microbiome analysis the cecum was not sampled at all? As SCFA are a functional parameter of the microbial community, there should be coherence in sampling sites for both. Otherwise no conclusions can be drawn.
Microbiota analysis: There is no information at all about the mode of sampling for faecal material. Furthermore, lines 523-527 attempt to give an idea about the aliquots taken per sampling site, but fail to describe it precisely. Does “specifically for microbiological purposes” actually means that content of the 5th Eppendorf tube sampled was used for sequencing? Please also provide information about the actual DNA-extraction/preparation prior to your sequencing.
Results:
The first presented data (lines 101-114, Table 1) on animal performance are already published in your previous paper (Zeebone et al., J Anim Physiol Anim Nutr. 2022;1–14, DOI: 10.1111/jpn.13724) and no reference is made here. You can certainly refer to the previous paper and give some information about animal performance but you cannot publish the same data twice (this isn’t good scientific conduct). Feed intake does not match you published data, although your statistical information is identical?
Serum biochemical measurements: You provide means±SD in Table 2 and if this is indeed the case I very much doubt your statistical p-value provided. Due to your huge standard variation, e.g. AST-values, there cannot be a significant difference of p = 0.01! Please revisit your statistical analysis and check this. This holds true for most of your parameters. Also, you provided a GFR (glomerular filtrate rate) as parameter and I was wondering how you obtained this? This is commonly a functional parameter and calculated using an inert marker such as inulin.
Antioxidant parameters: Please provide the actual data, at least as supplement. Also, there is no information on what tissues you were actually sampling and analysing for this in your M%M section.
Histopathology: Figure 1 supposedly shows micrographs of liver sections with HE staining – the resolution is absolutely too low thus the quality is insufficient. No information is provided on the actual magnification of these micrographs. Please provide a legible micrograph, otherwise it is no use to have this figure. Furthermore, I am somewhat amazed about the statement regarding glycogen content in hepatocytes (lines 158f) in HE stained tissue. A faint HE staining is certainly no proof of altered glycogen content as there are a number of methodological issues that might be the cause. There is a specific staining for glycogen reserves, the period-acid Schiff (PAS) technique, which is providing conclusive and objective data on this topic. Please remove these data on glycogen in HE staining as they are unsuitable for this purpose and provide instead either data obtained on PAS-stained liver tissue (including micrographs in proper resolution) or actual glycogen concentrations on liver tissue (also possible). Also, as mentioned above, provide a coherent objective scoring protocol for each organ/tissue.
SCFA & microbiota analysis: Please provide at least SCFA-data as supplement. Data presented on microbiota are somewhat disorganized and confusingly presented. Figure 2 gives a good impression on the phyla abundance in each gut section, but only for the control group. Why did you not provide your relative abundance of phyla for your other two groups? This would help the reader greatly in forming a picture on the microbiota. Also, it would be advisable to separate the end-point samples taken at slaughter (GIT content) from the repeated faecal samples – here as well it would be great to have one figure comprising the relative abundance on day 0 and day 21 of faecal material from all three groups. Please adjust these figures accordingly! Figures 3, 4, 5 should be also using the same unit as your initial Fig.2 or (%) or at least it would be more suitable for the reader to have coherent units in order to assess the actual shifts better.
This would also aid in a better understanding of the shifts in the microbiome you tried to present in Table 4. So far, this table presents data again in a very vague manner, actual numbers on phyla & genera in the GIT sections would be better suited to highlight the changes due to diets (alternatively distribution patterns per GIT section and diet group). In terms of statistical analysis, I’d recommend to revisit your methods: the GIT sections of each individual pig are not independent from each other, but rather a “repeated measure” and this should be included in the statistical model for these data set.
Discussion: Your discussion is fairly long and meanders on potential explanations for your microbiota changes, in particular with respect to function. This would indeed include information on microbial metabolites in the GIT sections and unfortunately none of your samples are taken with this aspect in mind. Sequencing the intestinal contents of these pigs without doing functional analysis (SCFA, including butyric acid; lactic acid, maybe ammonia; even just measuring pH after sampling) is very unfortunate and hinders you conclusions drawn from your presented data. If you do have still samples from the intestinal content fit for these analyses I’d strongly recommend these functional analyses.
